# A Pilot Study on Oxidative Stress during the Recovery Phase in Critical COVID-19 Patients in a Rehabilitation Facility: Potential Utility of the PAOT^®^ Technology for Assessing Total Anti-Oxidative Capacity

**DOI:** 10.3390/biomedicines11051308

**Published:** 2023-04-28

**Authors:** Joël Pincemail, Anne-Françoise Rousseau, Jean-François Kaux, Jean-Paul Cheramy-Bien, Christine Bruyère, Jeanine Prick, David Stern, Mouna-Messaouda Kaci, Benoît Maertens De Noordhout, Adelin Albert, Céline Eubelen, Caroline Le Goff, Benoît Misset, Etienne Cavalier, Corinne Charlier, Smail Meziane

**Affiliations:** 1Clinical Chemistry, University Hospital of Liège, Sart Tilman, 4000 Liège, Belgium; 2Intensive Care Department, University Hospital of Liège, Sart Tilman, 4000 Liège, Belgium; 3Physical Medicine Rehabilitation and Sports Traumatology Department Sports, University Hospital of Liège, Sart Tilman, 4000 Liège, Belgium; 4Department of Cardiovascular Surgery, University Hospital of Liège, Sart Tilman, 4000 Liège, Belgium; 5Veterinary Medicine Faculty, FARAH, University of Liège, Sart Tilman, 4000 Liège, Belgium; 6Research Department, Institut Européen des Antioxydants (IEA), Oxystress Technologies PAOTScan, 54500 Vandœuvre-lès-Nancy, France; 7Biostatistics Department, University Hospital of Liège, Sart Tilman, 4000 Liège, Belgium; 8Toxicology Department, University Hospital of Liège, Sart Tilman, 4000 Liège, Belgium

**Keywords:** post-COVID-19 pneumonia, patient rehabilitation, blood oxidative stress status

## Abstract

Background: Oxidative stress (OS) could cause various COVID-19 complications. Recently, we have developed the Pouvoir AntiOxydant Total (PAOT®) technology for reflecting the total antioxidant capacity (TAC) of biological samples. We aimed to investigate systemic oxidative stress status (OSS) and to evaluate the utility of PAOT® for assessing TAC during the recovery phase in critical COVID-19 patients in a rehabilitation facility. Materials and Methods: In a total of 12 critical COVID-19 patients in rehabilitation, 19 plasma OSS biomarkers were measured: antioxidants, TAC, trace elements, oxidative damage to lipids, and inflammatory biomarkers. TAC level was measured in plasma, saliva, skin, and urine, using PAOT and expressed as PAOT-Plasma, -Saliva, -Skin, and -Urine scores, respectively. Plasma OSS biomarker levels were compared with levels from previous studies on hospitalized COVID-19 patients and with the reference population. Correlations between four PAOT scores and plasma OSS biomarker levels were analyzed. Results: During the recovery phase, plasma levels in antioxidants (γ-tocopherol, β-carotene, total glutathione, vitamin C and thiol proteins) were significantly lower than reference intervals, whereas total hydroperoxides and myeloperoxidase (a marker of inflammation) were significantly higher. Copper negatively correlated with total hydroperoxides (r = 0.95, *p* = 0.001). A similar, deeply modified OSS was already observed in COVID-19 patients hospitalized in an intensive care unit. TAC evaluated in saliva, urine, and skin correlated negatively with copper and with plasma total hydroperoxides. To conclude, the systemic OSS, determined using a large number of biomarkers, was always significantly increased in cured COVID-19 patients during their recovery phase. The less costly evaluation of TAC using an electrochemical method could potentially represent a good alternative to the individual analysis of biomarkers linked to pro-oxidants.

## 1. Introduction

As initially defined by Sies [1], pathological oxidative stress (OS) has been initially defined as an imbalance between reactive oxygen species (ROS, such as free radicals, hydrogen peroxide, and singlet oxygen) and the antioxidant network in favor of the former, leading to irreversible oxidative damage to lipids, DNA and proteins. Oxidative damage is involved in the development of different pathologies including cancer and cardiovascular, neurodegenerative, and lung diseases [2]. However, molecular biology has highlighted that physiological ROS can act as secondary messengers, leading to the activation of important protective mechanisms for the body (e.g., apoptosis) [3]. To reconcile both pathological and physiological aspects of ROS, Jones [4] defined OS as an imbalance between ROS and antioxidants in favor of the former, leading to a signaling disruption as a consequence of irreversible oxidative damages to lipids, DNA and proteins. A third interesting and novel concept is adaptive oxidative stress or hormesis, defined as a phenomenon in which ROS produced in moderate amounts were able to stimulate antioxidant enzymes through activation of the Keap1/NrF2/ARE pathway [5].

In many diseases, inflammation is strongly related to pathological OS and *vice versa*. [6,7,8]. This is also true in COVID-19 pathogenesis, in which increased inflammation is associated with an elevated OS. This is a consequence of the inhibition of ACE-2 (angiotensin-converting enzyme 2), disseminated intravascular coagulation, the release of toxic free iron and of endothelial dysfunction [9,10,11,12]. In the most severe forms of COVID-19, pulmonary stress (fibrosis, loss of oxygenation capacity) and stress-linked sequelae of a cardiovascular (induction of pathologies or exacerbation of underlying chronic cardiovascular pathologies), renal (insufficiency), or cognitive and psychological nature (post-traumatic stress syndrome) have been reported after recovery from COVID-19, requiring medical follow-up and multidisciplinary rehabilitation [13,14,15,16,17,18]. These complications can be sustained by chronic inflammation and/or increased oxidative stress [19,20,21,22]. Numerous studies, including our own [23], have demonstrated the presence of significant pathological OS in COVID-19 patients hospitalized in an intensive care unit (ICU), characterized in particular by a collapse of plasma vitamin C and selenium levels and an increase in biomarkers of lipid peroxidation [24,25,26]. To correctly assess an oxidative stress status, it is necessary to use four classical axes of investigation: measurement of antioxidants and trace elements, evaluation of oxidized lipids, DNA, and proteins and, finally, identifying sources of ROS production (iron overload, inflammation, hyperglycemia, etc.). In the present study on COVID-19 patients during a recovery phase in rehabilitation facilities, we used 19 OS biomarkers belonging to these four categories.

Because these analyses take time and are costly, the total antioxidant capacity (TAC) in plasma has been proposed as a global measure of antioxidant deficiency [27]. The general principle of this method is to generate free radicals in a test tube of plasma and to measure their oxidative impact on different probes by spectrophotometry or chemiluminescence detection. After addition of a plasma sample to the medium, oxidative damage to the probe is expected to decrease because of the antioxidants present in the sample. Several spectrophotometric methods have been developed to measure plasma or serum TAC, such as Oxygen Radical Absorbance Capacity (ORAC), Total Radical-trapping Antioxidant Parameter (TRAP), and 2,2′-Azinobis-(3-Ethylbenzothiazolin-6-Sulfonic Acid (ABTS) [27], which should be more precisely renamed plasma or serum non-enzymatic antioxidant capacity (NEAC) [28]. A major drawback of these assays is that uric acid, the major antioxidant in plasma, reacts strongly with the free radicals used, thus masking the effects of other antioxidants such as vitamins C and E [29,30]. Recently, Carrion-Garcia et al. [28] also reported a strong and significant correlation (r = 0.67, *p* value 2.02 × 10^−19^) between uric acid and TAC, as assessed by the reduction of ferric iron (Fe^3+^) to ferrous iron (Fe^2+^) by antioxidants present in plasma samples (Ferric Reducing Antioxidant Power (FRAP)). Whether the TAC evaluated with such assays is related to plasma antioxidant concentrations remains a subject of debate.

To the best of our knowledge, there is no information about the short-term evolution of OS in severely ill COVID-19 patients related acute respiratory distress syndrome. The primary purpose of this prospective study was to determine OSS using a large battery of tests in COVID-19 patients hospitalized in a rehabilitation facility two months after their hospital discharge. The secondary objective was to evaluate the potential utility of PAOT^®^ technology for assessing TAC in plasma, saliva, urine and skin.

## 2. Materials and Methods

(a)
*Patient population*


The present prospective study was conducted in June 2020 on patients initially treated in ICU. We considered a convenient sample of 12 patients (aged ≥ 18 years) who recovered from sequelae in a rehabilitation facility after a COVID-19-related acute respiratory distress syndrome. Blood samples were drawn two months after hospital discharge. Demographic data were collected for all patients from the hospital electronic patient record, including medical history, ICU and hospital length of stay (LOS), critical illness severity score according to the Simplified Acute Physiology Score (SAPS) II [31], and organ supports and their durations (mechanical ventilation and renal replacement therapy). All patients were fed a standard hospital diet. No patient was under antioxidant supplementation. The study protocol was formally approved by the University of Liège hospital-faculty ethics committee (ethics code 707) under national reference B707202000035, local reference 2020-201, on 5 June 2020. All legal representatives of the patients were informed of the study objectives and signed informed consent. Data from the present study have been compared to a previous convenience sample (N = 9) of patients hospitalized for severe COVID-19 pneumonia, as published by our group [23].

(b)
*Assays for measuring OSS biomarkers*


To evaluate the oxidative stress status (OSS) of the study patients, four lines of research were followed: determination of both non-enzymatic (vitamin C, vitamin E as both α- and γ-tocopherol, β-carotene, total glutathione [tGSH], and thiol proteins [PSH]) and enzymatic antioxidants (glutathione peroxidase [GPx], determination of trace elements (selenium [Se], copper [Cu], and zinc [Zn]), determination of oxidation markers (total hydroperoxides [tROOH], oxidized low density proteins [ox-LDL], antibodies against oxidized LDL [IgG Ab-ox-LDL]), and identification of potential sources of increased OS production (the Cu/Zn ratio, C-reactive protein [CRP], myeloperoxidase [MPO], this last element being a specific biomarker of neutrophil activation).

The day before sampling, subjects fasted for 12 h and were not allowed to drink fruit juice. Between 8:00 and 9:00 a.m., blood samples were drawn from a venous central line into tubes containing, according to the investigated parameter, either an anticoagulant (EDTA or Na-heparin) or clot-activating gel. Blood samples were immediately centrifuged on site. Plasma or serum was then frozen as aliquots at −80 °C until analysis, which was performed within four days of blood collection. For vitamin C determination, 0.5 mL plasma was immediately transferred to tubes containing 0.5 mL of 10% metaphosphoric acid, and the whole mixture was frozen at −80 °C. Analyses were performed by a spectrophotometric method using reduction of 2,6-dichlorophenolindophenol (Perkin Elmer, Norwalk, CT, USA) [32]. Plasma vitamin E (α and γ-tocopherols) and β-carotene were determined simultaneously by HPLC (Alliance Waters, Washington, DC, USA) coupled to diode array detection (PDA 2996, Waters, Milford, MA, USA) [33]. The ratio of vitamin C to α-tocopherol was used as a potential risk factor for cardiovascular disease when decreased [34]. Thiol proteins were detected according to Ellman’s method [35] and glutathione peroxidase were measured in whole blood, respectively, with the GSH/GSSG-412 kit (Bioxytech, Oxis International, Inc., Portland, OR, USA) or the Ransel kit (Randox, RS SKU 504, Crumlin, UK). Plasma levels of selenium, copper, and zinc were determined by inductively coupled plasma mass spectrometry [36]. The analysis of total hydroperoxides (tROOH) as markers of oxidative damage was performed spectrophotometrically with a commercial kit (Oxystat, Biomedica Gruppe, Vienna, Austria). Oxidized low-density lipoprotein (LDL) in plasma samples was determined spectrophotometrically with a competitive enzyme-linked immunosorbent assay (ELISA) kit (Immunodiagnostik, Bensheim, Germany). The titer of free antibodies (IgG) against oxidized LDL (Ab-Ox-LDL) was assessed with a commercial enzyme immunoassay (Biomedica Gruppe, Vienna, Austria) using Cu^2+^ oxidized LDL as antigen. Myeloperoxidase (MPO) was assayed using a commercial ELISA kit (Immundiagnostik, Bensheim, Germany). CRP determination was analyzed by lumino-turbidimetry on an Alunity device (Abbott, Wavre, Belgium). The white blood cells count was determined on a Sysmex 9100 (Sysmex, La Hulpe, Belgium). All OS analyses are performed routinely in the central laboratory of the University Hospital of Liège, according the Clinical and Laboratory Standards Institute (CLSI) guidelines [37], and have been ISO 15189 accredited. In laboratory medicine, validation of methods using home methods or kits usually means analytical performance of methods (precision, linearity, carry-over, comparison, etc.). A minimum of 120 reference individuals are required for establishing 95% reference intervals with 90% confidence according to the CLSI EP23A3c guideline. According to this procedure, the reference intervals for each OS biomarkers have been in routine use and published in other papers of ours [38,39,40]. Individual concentrations of each OS biomarker in the two COVID-19 groups were compared to their reference intervals derived from the ELAN cohort study performed on 897 healthy subjects (349 men and 548 women).

(C)
*PAOT^®^-score determination*


PAOT^®^ (Pouvoir AntiOxydant Total)-Liquid, reflecting the total antioxidant capacity (TAC) of biological samples, was performed with electrochemical equipment as described in Figure 1A. The whole methodology has been previously described by us [41]. In a reaction medium (physiological solution at a pH ranging from 6.7 to 7.2 and temperature 24–27 °C) containing a free radical molecule (mediator M), two microelectrodes (the working and reference electrodes (patent FR2210844/PCT/FR2019/052835) were immersed. After addition of 20 μL of biological samples (plasma, urine, saliva), electrochemical potential modifications were recorded, resulting from changes in the concentrations of oxidized and reduced forms of the mediator M during reaction with antioxidants (AOX) present in the biological matrix (oxidized mediator M + AOX → reduced mediator M + oxidized AOX) [42]. Saliva samples were collected by Sarstedt-Salivette^®^ and by passive drooling into plastic tubes. After collection for 3 min, unstimulated saliva was centrifuged for 10 min at 3000× *g* and then stored at −80 °C until analysis. PAOT^®^ technology has also been adapted to evaluate in real time the redox equilibrium between antioxidants and oxidants in the skin [43], using the Oxystress skin analyzer (Figure 1B). Briefly, a patch consisting of 1 mL conductive gel containing both oxidized and reduced iron complex forms (mediator M) was applied to the arm skin areas. Then, working and reference microelectrodes coated with a four noble metal alloy (currently under patent WO2020/109736 A1) were connected to the patch. For the minutes that they were connected, the electrochemical potential shift was registered according to reactions between oxidized/reduced forms of mediator M with skin antioxidants (Pouvoir AntiOxydant Total or PAOT-Skin^®^) and oxidants (Pouvoir Oxydant Total or POT-Skin^®^), respectively. Finally, the PAOT-Skin Score^®^ was calculated as the ratio PAOT-Skin^®^/POT-Skin^®^ (for further detail see reference 43).

### Statistical Analysis

Data were presented as number (percentage) or median (min-max range). A sign test based on the binomial distribution was used to compare biological parameters measured in the COVID-19 patients with our laboratory reference intervals derived from the reference population. Specifically, if the 12 COVID-19 patients were not different from healthy subjects, we would expect approximately the same number of patients [44,45] above and below the middle of the reference interval. For instance, with five patients above the middle value (so-called positive patients) and seven below the middle value (so-called negative patients), it would be quite acceptable to say that the biological parameter does not differ between COVID-19 and healthy populations. By contrast, if all 12 COVID-19 patients fell below (all negatives) or above (all positives) the middle of the reference interval, then we could conclude that COVID-19 patients and healthy subjects differ for the biological parameter studied. Based on the binomial distribution and the assumption of no difference between the two populations (null hypothesis), the probability of such an extreme configuration happening is less than five in ten thousand (*p* < 0.0005). We reject the null hypothesis and conclude that COVID-19 and healthy populations are statistically different. Using the same argument, if 11 of the 12 COVID-19 patients are negative (or positive), the probability would be *p* = 0.006. According to the binomial distribution, probabilities are, respectively equal to *p* = 0.038 for 10/12 patients and *p* = 0.15 for 9/12 patients. Thus, there should be at least 10 of the 12 patients below (above) the middle of the reference interval to conclude that a significant difference exists. Results were considered significant at a 5% critical level (*p* < 0.05).

The Spearman correlation coefficient (r) was calculated to measure the association between biological parameters between them and with PAOT^®^-Scores observed in the different matrices. Significant correlation coefficients around r = 0.70 were considered “clinically relevant”, because the strength of association between the two parameters was approximately 50% (coefficient of determination r^2^ = 0.49) [46].

## 3. Results

As shown in Table 1, most patients presented with pathologies such as type 2 diabetes (42%) or arterial hypertension (75%), and/or were overweight. At the time of the blood test in June 2020, computed tomography of the chest was normal in five patients, while the others still showed partial regression. However, pulmonary infiltrates or fibrosis were only observed in three patients.

As seen in Table 2, the median concentrations of vitamin C, tGSH, PSH, γ-tocopherol, and β-carotene observed in our patients during their recovery phase were significantly lower than laboratory reference values. The median value of the PAOT^®^-score was also significantly lower but only in urine. By contrast, the median levels in tROOH, GPx, MPO, Cu/Zn ratio and CRP were significantly higher than reference intervals, while median values in copper, zinc, selenium, ox-LDL, Ab-ox-LDL and albumin were not significantly different. Of note, the median value of selenium was equal to the lower limit reference interval, while that of Ab-ox-LDL was higher than the upper limit but did not reach statistical significance. Appendix A display the distribution of individual OS biomarker values with respect to the reference interval in patients who recovered in the rehabilitation phase. Vitamin C (58.3%), PSH (66%) and GSH (91.6%) concentrations and the PAOT^®^-Urine Score (100%) were below the lower reference value (LRV) in almost all patients. GPx (100%), tROOH (83.3%), MPO (41.6%), CRP (75%) and IgG Ab-ox-LDL (100%) levels were also found at levels above the upper reference value (URV) in a large majority of patients. As shown in Table 2, similarly significant findings were already observed in COVID-19 patients during their stay in ICU. In this case, the median value of the tROOH and Cu/Zn ratio was higher than the reference interval; however, this was without reaching statistical significance. By contrast, albumin E was found to be significantly lower than the reference interval.

Table 3 shows the clinically relevant correlations observed between OS plasma biomarkers. tROOH correlated positively with copper (r = 0.95; *p* = 0.001) and to a lesser extent with the Cu/Zn ratio (r = 0.66, *p* = 0.020). In contrast, γ-tocopherol correlated negatively with tROOH (r = −0.61, *p* = 0.034) and copper (r = −0.66, *p* = 0.020). Vitamin C negatively correlates with ox-LDL (r = −0.72, *p* = 0.017). Table 3 also displays correlations between all PAOT^®^ Scores and OS biomarkers. Unlike the PAOT^®^-Plasma Score, the PAOT^®^-Saliva, -Urine and -Skin Scores correlated negatively and significantly with t ROOH and copper. By contrast, vitamin C positively correlated with PAOT^®^-Skin Score (r = 0.62, *p* = 0.043).

## 4. Discussion

In a previous pilot study [23], we evidenced an increased blood OS in nine COVID-19 patients hospitalized in an ICU for severe pneumonia. As compared to reference intervals, three major observations emerged: (1) severe and significant depletion of main antioxidants (tGSH and PSH, β-carotene, γ-tocopherol, vitamin C); (2) a significant correlation of tROOH with Cu and, to a lesser extent, the Cu/Zn ratio; (3) the presence of an inflammatory focus, as evidenced by increased levels of CRP and MPO (Table 2). The OS increase in hospitalized COVID-19 patients was later confirmed on other, larger populations of patients [47,48,49,50]. In fact, no information has been made available on patients having survived prolonged critical COVID-19 pneumonia and requiring lengthy rehabilitation after hospital discharge. In the present study, we highlight a persistent, deeply altered blood OSS profile during the convalescence of such patients, similar in most aspects to the profiles observed during an ICU stay [23].

*(a)* 
*Comparison between Plasma OSS Biomarker Levels in Study Population and the Previous Results*
*a.1.* 
*Antioxidants*



Besides being a crucial antioxidant as regenerator of both vitamins C and E, glutathione also plays a key role in good modulation of the immune system [51]. In both groups of patients, the GSH concentration was significantly decreased when compared to the reference interval.

Surprisingly and for unknown reasons, the blood concentration of GPx, requiring GSH as substrate for its antioxidant activity, was significantly above the reference interval in both groups (Table 2 and Appendix A). Given the elevated concentration in tROOH and the low levels of Se, this suggests probably that the GPx activity was not optimal. In parallel with tGSH, the large majority of patients exhibited low levels in PSH in both groups (Table 2 and Appendix A). Albumin represents about 70% of the PSH pool and can thus be considered an important antioxidant contributing up to 50% of the total antioxidant activity of plasma [52]. HSA-SH is expected to occur in human diseases and pathophysiological processes associated with increased oxidative stress [53]. If decreased concentration in PSH could partially be explained by low levels of albumin (28 g/L) in ICU patients [23], this is surprisingly not the case in recovered patients.

Eight isomers (the α-, β-, γ-, and δ-tocopherols and the α-, β-, γ-, and δ-tocotrienols-belong to the vitamin E family. The present study shows that γ-tocopherol levels, but not α-tocopherol levels, were significantly shifted downward in both groups of patients (Table 2 and Appendix A). Through its antioxidant activity and also its regulation of various enzymatic pathways, γ-tocopherol is an important antioxidant because it reduces the risk of cardiovascular diseases and cancer [54].

As shown in Appendix A, 58.3% of patients had hypovitaminosis C (value < 6 µg/mL), as previously defined by Lindblat et al. [55]. Although COVID-19 patients admitted to the ICU also had low vitamin C levels [23,24], persistence of vitamin C depletion (Table 2) two months after hospital discharge was intriguing, as the patients received standardized nutritional vitamin C intakes during their recovery in rehabilitation facility. The recent prospective PRIME study performed on 9709 men aged 50–50 years showed that low plasma levels in vitamin C were associated with coronary events [56]. Gey et al. [33] also reported that ideally, to offer a maximal cardio-protective effect, the vitamin C/vitamin E (α-tocopherol) ratio (reflecting synergy between the two molecules) must be higher than 1.3–1.5 when the concentrations of both vitamins were expressed in µM. In contrast, a ratio as low as 0.6 was associated with a higher risk of cardiovascular diseases. In our study, 66% of the patients during their recovery phase displayed a ratio below 1.3 (data not shown). With respect to β-carotene, a median concentration of 0.22 µg/mL (range 0.18–0.22 µg/mL) was found quite close to the LRV (Table 2). Such a concentration is, however, not optimal in terms of health prevention because it is associated with the development of cardiovascular diseases and cancer [33].

Such non-enzymatic antioxidant depletion in cured COVID-19 patients raises the question of potential correction through either optimized nutrition or supplementation. A daily intake of five portions of fruits or vegetables [57] or a daily supplement of vitamin C (100 µg) [58] allows rises in vitamin C levels (by about 10 µg/mL) in populations having initials levels around 5 µg/mL as observed in our study. Interestingly, it has been shown that a preparation of mixed tocopherols containing γ-, δ-, and α-tocopherol (5:2:1), as found in corn and soybean oils, has higher antioxidant and anti-inflammatory activities than α-tocopherol alone [59,60]. Improvement in glutathione concentration could be achieved by the ingestion of whey proteins in the form of daily consumption of 250 mL milk containing the A2 type of β-casein for two weeks [61]. As a precursor of glutathione, N-acetyl-L-cysteine (600 mg) has also been proposed [62].

*a.2.* 
*Trace Elements*


Se is known to play a key role in initiating immunity and in regulating chronic inflammation or an excessive immune response [63]. Similarly, to COVID-19 patients in intensive care [23,25], the studied patients admitted to the rehabilitation facility exhibited a median concentration very close to the LRV of 73 µg/L (Table 2). Bomer et al. [64] reported that a blood concentration ≤70 µg/L, as observed in 55% and 25% of hospitalized and recovered patients, respectively, was associated with symptoms of heart failure, poorer exercise capacity and all-cause mortality. If plasma Se is below 89 µg/L, it is necessary to increase selenium intake through diet or supplementation (70 µg) in order to reach an ideal plasma selenium of around 122 µg/L, as recommended by Steinbrenner et al. [65].

Copper exhibits pro-oxidant activity at non-physiological concentrations through the Fenton reaction [66]. In our study, the pro-oxidant effect of copper was suggested by its positive correlation with tROOH and its negative correlation with γ-tocopherol (Table 3). In contrast, zinc plays an important role in immunity and also exhibits antioxidant properties, notably by inhibiting the free radical reaction induced by copper [67]. Although the plasma concentrations of copper and zinc were within their reference ranges, our patients showed a Cu/Zn ratio significantly higher than the reference interval in patients recovering in rehabilitation phase (Table 2). Moreover, this ratio correlated positively with tROOH (Table 3), in accordance with other studies [68,69,70]. Interestingly, supplementation with zinc as a gluconate (78 mg/day during 8 weeks) has been reported to reduce Cu/Zn ratio concomitantly with lipid peroxidation in hemodialysis patients [71].

*a.3.* 
*Oxidative Damage to Lipids*


Increased ROS production leads to oxidation of important biological substrates such as amino acids, peptides, proteins [72], and lipids (particularly polyunsaturated fatty acids) [73], resulting in the appearance of the hydroperoxide (-OOH) function. With the Oxystat kit (Biomedica) commonly used in many studies, we evidenced high tROOH concentrations in both groups (66.6% in ICU and 83.3% during rehabilitation; Appendix A) and more particularly in a significant way in the second group. This could not be attributed to hydrogen peroxide (H_2_O_2_), since its concentration in plasma is very low due to catalase activity [74]. However, some papers reported that plasma ceruloplasmin was able to catalyze the oxidation of cyclic hydroxylamine probes as trimethybenzidine (TMB), used for detecting -OOH function due to its peroxidase activity [75]. This should therefore mask the real level of tROOH. If ceruloplasmine is well known to act as an antioxidant through its ferroxidase activity, it also exhibits prooxidant properties via amino oxidase and NO oxidase activities [76]. Moreover, it has been shown *in vitro* that purified ceruloplasmin was able to increase LDL oxidation [77]. *In vivo* studies have also evidenced that ceruloplasmin significantly and positively correlated with malonaldehyde (MDA), as another marker of lipid peroxidation, in the plasma of rheumatoid arthritis patients [78]. It is well accepted that increased lipid peroxidation is clearly involved with the development of atherosclerosis and cardiovascular diseases [79]. In a study performed on 123 healthy individuals, Miller III et al. [80] showed that a combination diet rich in fruits and vegetables consumed for three weeks protects against lipid peroxidation when compared to a control diet.

By contrast, levels of ox-LDL remain within the normal reference interval in both groups. However, ox-LDL are well known to overexpress the production of both IgM and IgG ox-LDL antibodies. If IgM antibodies have anti-atherogenic properties, IgG antibodies are, in contrast, characterized by potent pro-inflammatory activity [81]. By contrast to ICU patients, we found high titers in IgG Ab-ox-LDL in all partially or totally recovered patients (Appendix A). This is particularly intriguing since it has been shown that high levels in IgG anti-oxLDL titers were associated with the extent of atherosclerosis and cardiovascular diseases [82]. This biomarker could therefore be useful to measure during the recovery phase of COVID patients. Indeed, Xie et al. [83] highlighted in a cohort of 153,760 US veterans who survived the first 30 days of COVID-19 that the risk of cardiovascular events is significant within 12 months of SARS-CoV-2 infection.

*a.4.* 
*Inflammation Biomarkers*


Lastly, the patients admitted in the rehabilitation facility exhibited, like those in ICU, an inflammatory terrain, as evidenced by a median plasma MPO and CRP concentrations slightly but significantly higher than the reference interval (Table 2). As shown in Appendix A, the MPO concentration could reach very high values (>100 ng/mL) similar to those observed in COVID-19 patients during their stay in the ICU [23]. Such an inflammatory process as a major source of ROS production could be partially responsible for the depletion of the antioxidant defenses observed in our study. Moreover, it is well known that MPO can in the long term drive the development of numerous chronic inflammatory pathologies responsible for increased patient mortality and morbidity [84].

*(b)* 
*Correlations between OS Plasma Biomarkers*


Table 3 shows that Cu was highly and positively associated with tROOH, the clinical relevance being significate. Such observations have been reported in other clinical studies associated with increased OS [85]. Of interest was to note the inverse and significant correlation between γ-tocopherol and tROOH. At last, vitamin C also negatively and highly correlates with ox-LDL whose median value is very close to the upper reference interval. Both observations suggest that a depletion in antioxidants can effectively contribute to increasing lipid oxidation.

*(c)* 
*PAOT^®^-Scores and Correlations with OS Plasma Biomarkers*


In a second part of the study we also examined whether the use of PAOT^®^ technology to determine TAC might be useful as part of OSS evaluation in both group of COVID-19 patients. For this, we used four different biological matrices. As shown in Appendix A, the PAOT^®^-Plasma Score was moderately shifted downwards in recovered patients. However, its median was not statistically different form the reference interval (Table 2) despite significant decreases in plasma vitamin C, GSH, and PSH. Moreover, no correlation was found between the PAOT^®^-Plasma Score and plasma antioxidants. This was most probably due to the well-known interference of uric acid, as reported for other TAC assays [27,28]. In ICU patients, we found, however, a significant decrease in the PAOT^®^-Plasma Score (Table 2). This discrepancy could be potentially related to median value of albumin being lower in ICU patients (28 g/L, *p* = 0.040) than the reference inference by comparison to patients in rehabilitation phase (38 g/L, *p* = 1.0). As said above, albumin represents by itself between 10 to 50% of the total antioxidant capacity of plasma.

Urine is considered as a useful biological matrix for routine testing of inflammatory [86], OS biomarkers [87] and TAC [88]. As shown in Appendix A, all individual PAOT^®^-Urine Scores were well below the LRI. Despite this, we found no correlation with antioxidant plasma concentrations. In contrast, we evidenced significant, relevant and negative correlations between the PAOT^®^-Urine Score and both plasma tROOH (a marker of oxidation) and Cu (known to catalyze ROS production and to induce lipid peroxidation) concentrations (Table 3).

In clinical studies, it has also been proposed to determine TAC in saliva concentrations, because of its availability and non-invasive collection [89]. Besides oxidized lipids and proteins [90], other antioxidants such as vitamin C or GSH [91] have been detected in saliva, as well as oxidants such as hydrogen peroxide [92] and also hypochlorous acid (HOCl) [93] resulting from increased MPO activity. Therefore, saliva thus appears as an excellent candidate matrix for evaluating the general redox status (balance between oxidants and antioxidants) of an individual. It is important to note, however, that results may be affected by dental hygiene or local oral infection [94]. In our study, the majority of individual PAOT-Saliva Scores were low as compared to the reference interval (Appendix A). Moreover, Table 3 shows that the PAOT-Saliva Score also correlates negatively with plasma tROOH and copper concentrations (Table 3), with a clinical relevance.

With the largest surface area in the body, the skin is a promising way to evidence redox status [95] as it is continually exposed to oxidizing attacks from both inside and outside the body [96,97]. Brainina’s group [98] was the first to determine both antioxidant (AOA) and oxidant (OA) activities in the skin, using a K_3_[Fe(CN)_6_/]/K_4_[Fe(CN)_6_] mixture as a reduced/oxidized mediator (M) system, platinum as the working electrode, and electrocardiogram (ECG) electrodes as the reference. The great advantage of such an approach is to measure in real time the redox status of the skin. Furthermore, this methodology is easy to use and is not time-consuming. With PAOT^®^ technology, which is similar but uses more sensitive electrodes, we have evidenced negative correlations with a clinical relevance between the PAOT^®^-Skin Score and both plasma tROOH and Cu concentrations (Table 3). These findings are in agreement with our previous study on 30 healthy subjects [43]. Interestingly, we also found a positive and significant association (r = 0.62, *p* = 0.043) between plasma vitamin C and PAOT^®^-Skin Score, even if not being clinically relevant.

Overall, the present study on COVID-19 patients highlights some weaknesses of TAC determination as a means of directly assessing the antioxidant status of a patient, whatever the biological sample (plasma, urine, saliva, skin). On the other hand, PAOT^®^ technology, particularly applied to samples collected non-invasively, could prove useful in detecting a pro-oxidant state in pathological situations, as evidenced by negative correlations with a clinical relevance of PAOT^®^-Saliva and Skin Scores with Cu and tROOH. Further study is imperative in order to strengthen these conclusions.

## 5. Limitations of the Study

First, for practical reasons (notably the high cost of OS assessments), we have studied only 12 patients. Second, some of them presented medical conditions (diabetes, arterial hypertension) potentially associated with OS. If focusing on vitamin C, these pathological situations by themselves were, however, never associated with hypovitaminosis C (<6 µg/mL) For example, Sinclair et al. [99] described the following respective vitamin C concentrations in a control group with normal glucose tolerance, in prediabetes patients, and in diabetes patients: 10.1 µg/mL, 8.54 µg/mL, and 7.29 µg/mL, with all values being in the reference interval. In a meta-analysis of observational studies on 5271 hypertensive patients, Ran et al. [100] concluded that the median concentration in vitamin C was 7.4 µg/mL (min: 4.89–max: 8.73), which is largely higher than those observed in our study.

## 6. Conclusions

Using a large battery of tests, we have confirmed a significantly and highly disturbed oxidative stress profile systemic OSS profile in convalescent survivors of critical COVID-19. All the anomalies observed here are well recognized as being associated in the long term with the development of human pathologies, particularly cardiovascular diseases. Whether the antioxidant capacity determined with PAOT^®^ technology alone might be used as a surrogate for increased OS remains a challenging question, but our preliminary results seem to indicate that PAOT^®^ Score in saliva, urine, and skin could be useful in evidencing a pro-oxidant activity rather an antioxidant depletion. All the observations presented in this work allow us to potentially ask to what extent an increase in antioxidant defenses, either by an appropriate diet or by supplementation, would be useful in COVID-19 patients during their recovery period.

## Figures and Tables

**Figure 1 biomedicines-11-01308-f001:**
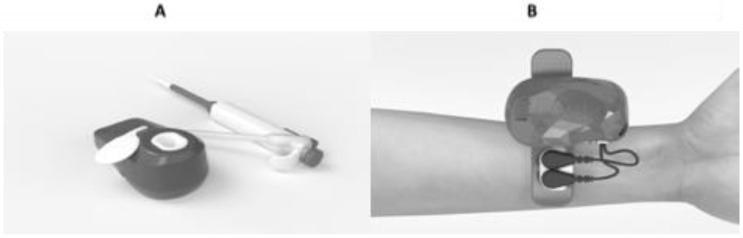
Photograph of the PAOT^®^-Liquid device for TAC evaluation in plasma, saliva, and urine (**A**) and of the Oxystress skin analyzer (**B**) for determination of the PAOT^®^-Skin Score Pictures kindly provided by IEA.

**Table 1 biomedicines-11-01308-t001:** Demographic data of the 12 COVID-19 patients upon admission to ICU for severe pneumonia (N = 12). BMI: body mass index; LOS: length of stay; SAPS II: simplified acute physiology score II. Data are presented as median (min-max range) or number (percentage). * Clinical observations made at the time of blood collection during recovery in rehabilitation facility.

Variable	Median (Range)
Age (y)	64 (47–76)
Male gender	9 (75)
Weight (kg)	94 (50–118)
Height (cm)	177 (152–196)
BMI (kg/m2)	29.8 (21.9–33.9)
Active smoking	0 (0)
Active alcoholism	0 (0)
Pre-existing medical conditions	
-Type 2 diabetes	5 (42)
-Arterial hypertension	9 (75)
-Hypothyroidism	4 (33)
SAPS II	33 (20–82)
ICU LOS (d)	42 (15–57)
Hospital LOS (d)	53 (41–138)
Mechanical ventilation duration (d)	27 (8–50)
CVVH, n (%)	2 (17)
Complete regression *	5(41.6)
Partial regression *	7 (58.3)
Presence of fibrosis *	3 (25)

**Table 2 biomedicines-11-01308-t002:** Median concentration (min-max range) of plasma OS biomarkers in COVID-19 patients hospitalized in ICU and those who recovered from COVID-19 (N = 12) in rehabilitation facilities two months after their hospital discharge. Statistical comparison (*p*) with the middle of reference intervals was performed using the sign test [23]. * Median (range) and *p*-values are from reference 23.

	COVID-Patients in ICU (N = 9)		COVID-19 Patients in Rehabilitation Facility (N = 12)		
Parameter	Median (Range)	*p* Value *	Median (Range)	*p* Value	Reference Interval (N=897) [38,39,40]
*Antioxidants*					
Vitamin C (µg/mL)	3.91 (3.06–6.14)	0.004	5.43 (2.51–10.90)	0.006	6.0–18
Vitamin E as α-tocopherol (µg/mL)	17.90 (13.3–21.1)	1.0	17.3 (11.1–26.6)	1.0	8.6–19.2
γ-tocopherol (µg/mL)	0.84 (0.57–1.28)	0.040	0.95 (0.39–2.60)	0.006	0.39–2.42
β-carotene (µg/mL)	0.14 (0.06–0.68)	0.004	0.22 (0.09–0.0.40)	0.006	0.06–0.68
Thiol proteins (PSH) (µM)	250 (204–258)	0.004	242 (197–357)	<0.0005	314–516
Total glutathione (tGSH) (µM)	629 (508–697)	0.040	614 (450–721)	<0.0005	717–1110
Oxidized glutathione (GSSG) (µM)	<0.96	1.0	<0.96	1.0	0.96–10
PAOT^®^-Skin Score			67.7 (11.5–115)	1.0	7.86–62.9
PAOT^®^-Plasma Score	10.52 (6.63–10.77)	0.04	29.4 (12.2–79.2)	1.0	1.42–36.78
PAOT^®^-Saliva Score			7.03 (0.88–22.1)	1.0	1.52–14.14
PAOT^®^-Urinary Score			11.8 (8.6–26.4)	<0.0005	43–105
Glutathione peroxidase (GPx) (UI/g Hb)	69.55 (61.9–78.27)	0.004	78 (51–103)	<0.0005	20–56
Albumin (g/L)	28 (27.5–33.0)	0.04	38 (32–44)	1.0	32–46
*Trace elements*					
Copper (Cu) (mg/mL)	1.16 (0.66–1.47)	1.0	1.10 (0.76–1.56)	0.15	0.70–1.1
Zinc (Zn) (mg/mL)	0.84 (0.81–1.01)	1.0	0.95 (0.56–1.18)	1.0	0.70–1.20
Selenium (Se) (µg/mL)	74 (59–103)	1.0	76.3 (52.1–102)	0.39	73–110
*Biomarkers of oxidation*					
Total hydroperoxides (tROOH) (µM)	674 (181–1415)	0.50	1061 (162–1905)	0.006	0–432
Oxidized LDL (ox-LDL) (ng/mL)	50 (36–70)	1.0	67 (31–114)	0.39	28–70
Antibodies against oxidized LDL (Ig G Ab-ox-LDL) (IU/L)	306 (64–1200)	1.0	870 (70–8190)	0.39	200–600
*Inflammatory biomarkers*					
Copper/zinc ratio (Cu/Zn)	1.55 (0.79–1.69)	1.0	1.24 (0.74–1.68)	0.038	1–1.17
Myeloperoxidase (MPO) (ng/mL)	88 (60–191)	0.04	72 (40–124)	0.038	27–72
C-reactive protein (CRP) (mg/L)	32.8 (9.6–59.8)	0.04	10.2 (1.2–45.4)	0.038	0–5
White blood cells 10^3^/mm^3^	8.42 (7.07–13.03)	0.04	6.9 (3.0–9.0)	1.0	4.6–10.1

**Table 3 biomedicines-11-01308-t003:** Correlations between OS plasma biomarkers between them and PAOT^®^-Urine, -Saliva, -Skin and Plasma Scores, as observed in patients sent to a rehabilitation center following COVID-19 disease (N = 12). tROOH: total hydroperoxides.

Parameter	Parameter	Spearman Correlation	Determination Coefficient	*p*-Value
Cu	Cu/Zn	0.63	0.39	0.028
Cu	γ-tocopherol	−0.66	0.43	0.020
Cu	tROOH	0.95	0.90	0.001
Cu/Zn	tROOH	0.66	0.43	0.020
γ-tocopherol	ROOH	−0.61	0.37	0.034
ox-LDL	vitamin C	−0.72	0.51	0.017
PAOT^®^-Plasma Score	selenium	0.68	0.46	0.015
PAOT^®^-Urinary Score	Cu	−0.72	0.51	0.008
PAOT^®^-Urinary Score	ROOH	−0.69	0.47	0.013
PAOT^®^-Saliva Score	ROOH	−0.78	0.60	0.005
PAOT^®^-Saliva Score	Cu	−0.81	0.65	0.003
PAOT^®^-Saliva Score	Cu/Zn	−0.59	0.34	0.055
PAOT^®^-Saliva Score	γ-tocopherol	0.59	0.34	0.055
PAOT^®^-Skin Score	ROOH	−0.77	0.59	0.005
PAOT^®^-Skin Score	Cu	−0.76	0.57	0.007
PAOT^®^-Skin Score	vitamin C	0.62	0.38	0.043

## Data Availability

The datasets analyzed during the current study are available from the corresponding author on reasonable request.

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
