# Peer review of "A Pilot Study on Oxidative Stress during the Recovery Phase in Critical COVID-19 Patients in a Rehabilitation Facility: Potential Utility of the PAOT® Technology for Assessing Total Anti-Oxidative Capacity"

_biomedicines, 2023, doi:10.3390/biomedicines11051308_

Round 1

Reviewer 1 Report

It is not a well written paper due to very poor English quality.   Throughout the paper, English expressions are vague and not clear.  1) In the abstract, what does standards mean? Do you have controls? There are no sentences about controls in the Materials and Methods section.        : Vitamin C, thiol proteins, reduced glutathione, r- 35 tocopherol and b−carotene were significantly decreased compared to standards.   2) The use of the term "altered" is not inappropriate. Authors did not measure blood OSS consecutively in the study patients.      Conclusions: Systemic OSS was strongly altered in survivors of a critical COVID-19 pneumonia. This suggests the need for supplementing these patients with antioxidants     The authors have repeatedly described the purpose of this study in both the introduction and Materials and Methods sections,  It is questionable whether the study population was selected for the purpose.   What is the purpose and main findings in this study? In addition, there is no information on the enrollment/exclusion criteria of the study population. What is the study period? What is the definition of critical COVID-19 pneumonia?  Authors measured blood OS biomarker levels in recovered COVID-19 patients without providing the information on the status of study population. Did the study population have complications such as post-COVID-19 fibrosis?  Were there any evidence of chronic inflammation in the study population? (e.g., remained symptoms or pneumonia, radiologic abnormalities, increased inflammatory biomarkers like WBC, CRP, or procalcitonin)   Also, I could not understand the meaning of Table 1.  Are the reference intervals in Table 1 the manufacturer-claimed reference intervals? What does the p-value mean in Table 1?  Authors should present median (interquartile range) in Table 1. The use of the figure would be helpful to show the distribution of blood OS biomarker levels   It is uncertain whether blood OS biomarker levels reflect inflammation due to COVID-19 or other underlying diseases.  Therefore, authors should be cautious about suggesting regular check of OSS biomarkers in COVID-19 convalescence could be recommended to carry out antioxidant supplementation.

Reviewer 2 Report

Good morning

for the Authors,

Analyzing the Manuscript (Article) with ID: biomedicines-2124990-peer-review-v1, entitled "Determination of Redox Status in surviving COVID-19 patients during their recovery in a rehabilitation facility. Potential use of the electrochemical PAOT® technology. A Pilot Study" for a possible publication in Journal Biomedicines – MDPI (ISSN: 2227–9059; IF=4.757), Section: Immunology and Immunotherapy, Special Issue: Advanced Biomedical Research on COVID-19;

I consider that:

1. The authors of the article proposed a much-discussed topic in the medical scientific world today, namely: the recovery of post-COVID-19 patients, especially those who had serious/critical clinical manifestations and who were hospitalized in the wards UTI. The effort made by the authors was great right from the beginning of the study.

2. In Chapter 1 – Introduction (well structured) the authors presented the reasons for choosing their study which are congruent with those of other authors cited as references.

3. In Chapter 2 – Materials and Methods:

- The authors clearly presented the inclusion criteria of patients in their study: demographic data, duration and type of study, for a representative cohort of patients who were previously hospitalized in UTI departments (n=12).

- The authors presented the consent of the ethics committee of the institution where they conducted the study. They had an informed, written consent of the patients in the study, thus respecting all the provisions of professional ethics.

- They analyzed a very large number of biomarkers of the systemic oxidative stress status (OSS) - enzymatic and non-enzymatic antioxidants, markers total antioxidant capacity of plasma (PAOT-Score®), trace elements, marker oxidative damage of lipids and inflammation markers.

- They use a standard protocol for collecting, processing and preserving the biological samples of patients.

- To evaluate the levels of these markers, they used laboratory techniques and kits with high-performance technological, respecting the provisions of the guidelines approved by the IFCC and the ISO-15189 norms.

- The reference values for the analyzed markers were those presented in well-documented scientific studies (the ELAN cohort study) and by other authors specified in the references.

- The authors clearly and concisely presented the usefulness of the PAOT® (Pouvoir AntiOxydant Total)-Liquid electrochemical equipment for evaluating the total antioxidant capacity (TAC) (presented in Figure 1).

- The authors used specific calculation formulas that are well presented. Other results obtained by the authors (especially the first author) by using the PAOT® equipment were published in prestigious journals (8 articles, specified in the references).

- The authors also have an approved and validated invention patent for this equipment.

- To process the obtained data, the authors used different tests and specific statistical analysis coefficients (p and r value). They had a significant value in assessing the friability of the study.

4. In Chapter 3 – Results:

- The authors presented clearly and concisely in Table 1 the demographic data and other pathological parameters of the patients in the study.

- In Table 1, the authors coherently presented the levels of the analyzed markers compared to the reference values, specifying for each parameter the value of the statistical coefficient, p value.

- In Figure 2 and biomedicines-2124990-supplementary, the authors clearly presented the correlations between the levels of the analyzed markers, specifying for each parameter the value of the statistical coefficients: p and r value.

5. In Chapter 4 – Discussions:

- The authors clearly presented in Table 3 - Correlations between PAOT-Urine, -Saliva and -Skin Scores® and the plasma levels of the biomarkers analyzed in the patients in the study.

- The authors compared their results with those of other authors according to the bibliography/references.

6. In Chapter 5 – Limitations of the study: autorii prezinta argumentele practice pentru folosirea unui numar mic de pacienti (n=12), dovedint si onestitate stiintifica.

6. In Chapter 6 – Conclusions:

- The authors clearly and concisely presented the involvement of oxidative stress markers in the serious clinical manifestation of patients from the Covid-19 pandemic.

- The authors presented the great importance of the PAOT® technology in evaluating the total antioxidant capacity (TAC).

7. All authors had a fair contribution in the realization of the study.

As a result:

1. The article follows all the specific instructions of the journal presented in: aims and scope, instructions for authors and other information about the journal.

2. The data presented in this manuscript are well structured and coherent;

3. The methods, statistical analysis and results are well presented and easy to understand;

4. The references chosen by the authors corresponds to the requirements and refers to the subject of the article.

5. All authors had an equitable contribution to the study.

 In conclusion:

I Accept in present form!

Round 2

Reviewer 1 Report

Thank you for your revision. Unfortunately, despite your best efforts, the manuscript is not well organized. 

1. The authors compared the plasma OSS biomarker levels of 12 patients with previously reported levels from a reference population. The study design is very similar to your previous study, Pincemail, et al.  The format of Table 2 is almost same with Table 4 in Pincemail, et al. This design and format seem not to be original and novel. For the originality and novelty of this study, I suggest that results of this study need to be interpreted by comparing both the result oPincemail, et al and previously reported results from a reference population. Modify Table 2 to include more columns for present study,  Pincemail, et al, and reference population.  It seems sufficient to explain trends of OSS biomarker levels during recovery phase by comparing them with two previous studies without statistical analysis for comparison.

Reference of format: Schüring, Andreas N et al. “Establishing reference intervals for sex hormones on the analytical platforms Advia Centaur and Immulite 2000XP.” Annals of laboratory medicine vol. 36,1 (2016): 55-9. doi:10.3343/alm.2016.36.1.55

2. I suggest changing the title as follows;

A pilot study on the oxidative stress status during recovery phase in critical COVID-19 patients in rehabilitation: Utility of the PAOT technology for assessing total oxidative capacity

3. Lines5-7: Add a superscipt "*" to the corresponding author's name.

4. The abstract is not well organized. 

1) Background

Describe the only main background and purpose.

(Example) Oxidative stress (OS) could cause various COVID-19  complications. Recently, Pouvoir AntiOxydant Total (PAOT) technology has been developed to measure total oxidative capacity (TAC) reflecting OS status (OSS). We aimed to investigate plasma OSS biomarkers and to evaluate the utility of PAOT for assessing TAC during recovery phase in critical COVID-19 patients in rehabilitation.

2) Materials and Methods

Describe the materials and methods briefly. Check the number of total plasma biomarkers and describe it.

(Example) In a total of 12 critical COVID-19 patients in rehabilitation, 19 plasma OSS biomarkers were measured; antioxidants, TAC, trace elements, oxidative damage to lipids, and inflammatory biomarkers. TAC level was measured in plasma, saliva, skin, and urine using PAOT and expressed as PAOT-Plasma, -Saliva, -Skin, and -Urine scores, respectively. Plasma OSS biomarker levels were compared with levels from previous studies on hospitalized COVID-19 patients and reference population. Correlation between four PAOT scores and plasma OSS biomarker levels were analyzed.

3) Results & conclusions

Describe them according to results of new analyses.

5. Describe the introduction section logically.

The definition and types of OS --> the relationship between OS and COVID-19 -->Pathogenesis of OS in COVID-19 complications--> the previously reported results of OSS in COVID-19--> limitations of OS measurement --> descriptions of TAC and PAOT technology--> To the best of our knowledge, no studies evaluated about ~~ --> purpose. 

6. Line 45: What does the medium-term evolution mean?

7. Lines 64-66

1) Line 66: Did you use serum or plasma for TAC? Please check it.

2) Modify these sentences.

(Example) The first purpose of this study was to investigate plasma OSS during recovery phase in critical COVID-19 patients in rehabilitation using a large battery of tests. The second objective was to evaluate the utility of PAOT technology for assessing TAC in plasma, saliva, skin, and urine.

8. Line 68 ~:

The Materials and Methods are not well organized. Add "Study Populaltion, and "Assays for measuring OSS biomarkers" as subheadings, and describe according to subheadings.

9. Line 69: The study--> The prospective study

10. Line 70: critical COVID-19 pneumonia (severe~)

In WHO guideline, critical COVID-19 is defined as ARDS, sepsis, septic shock, or acute thrombosis. Did all patients have ARDS?

Reference: Clinical management of COVID-19 - Living guideline (13 January 2023). https://www.who.int/teams/health-care-readiness/covid-19

11. Line 71: Intensive Care Unit-->ICU

12. Line 72: (BMI)-->[BMI]. Check the parenthesis throughout the manuscript.

13. Line 72-77

1) critical illness severity score

What score was used for  COVID-19 disease severity? SAPSII in Table 1? Describe it in detail with reference in this section. 

2) What do organ supports mean? mechanical ventilation in Table 1? Please use clear terms.

3) At the time of the blood test in 74 June 2020, two months after hospital discharge from hospital, the computed tomography of the chest was normalized in 4 patients, while the others still showed pulmonary infiltrates or fibrosis.--> How many are fibrotic patients? add N (%) of CT results in detail in Table 1.

4) C reactive protein blood levels were slightly higher than reference standards (0 – 5 mg/L), reaching 10.2 (min: 1.2 – Max: 45.4), but white blood cell counts were within reference standards 4.6 10.1.10.3/mm3 with a median value of 6.9 (min: 3 –max:3-8.96)---> These sentences are results. Add median (IQR) levels of CRP, WBC levels to Table 1. The term of reference standard is not appropriate. The term of reference intervals is correct. In addition, these sentences are wrong. Was median CRP level of the study population higher than upper reference limit (5 mg/L)? or were most CRP levels of the study population  higher than upper reference limit (5 mg/L)? 

5) Some patients showed abnormal CT results and high CRP levels. I'm not sure if they are recovering. 

14. Line 104-109:

1) All OS analyses performed routinely in clinical practice by the Central Laboratory of Liège CHU according the Clinical and Laboratory Standards Institute (CLSI) guidelines were ISO 15189 accredited--> Add reference.

2) To validate home-methods or those using a kit, a minimum of 120 reference  individuals were required according to the International federation of clinical chemistry (IFCC). 

--> In laboratory medicine, validation of methods usually means analytical performance evaluation of methods (precision, linearity, carry-over, comparison, etc). Minimum 120 reference individuals are required for establishing reference intervals with 90% confidence intervals according to the CLSI EP28A3c guideline. If all analytical performance-evaluated assays were used in this study, describe it briefly.

15. Statistical analysis

Describe statistical analysis in the order in which you analyzed them according to your purpose.

1) Comparison between plasma OSS biomarker levels in study population and the previous results.

2) Correlation among plasma OSS biomarkers

3) Correlation between PAOT scores and plasma OSS biomarkers.

4) Interpretation of correlation coefficient with reference

Reference: Mukaka MM. A guide to appropriate use of correlation coefficient in medical research. Malawi Med J 2012;24:69–71.

16. Line 129:

1) Quantitative OS data were expressed as median and range while numbers (percent) were used for categorical data.--> Data were presented as number (percentage) or median (interquartile range [IQR]). 

2) Spearman correlation: 

Pearson correlation analysis is used for continuous variables (e.g., body weight, height, etc.). Spearman correlation analysis is used for rank-ordered variables such as grades or levels. Has a statistician been consulted about correlation methods?

Round 3

Reviewer 1 Report

The draft should be revised extensively .

Please reflect my suggestions especially for statistical analysis.

I hope that authors will revise the draft with their best.
